# MicroRNA Profile, Putative Diagnostic Biomarkers and RNA-Based Therapies in the Inherited Lipid Storage Disease Niemann-Pick Type C

**DOI:** 10.3390/biomedicines11102615

**Published:** 2023-09-23

**Authors:** Marisa Encarnação, Hugo David, Maria Francisca Coutinho, Luciana Moreira, Sandra Alves

**Affiliations:** 1Research and Development Unit, Department of Human Genetics, National Institute of Health Doutor Ricardo Jorge, INSA I.P., Rua Alexandre Herculano 321, 4000-055 Porto, Portugal; hugo.david@insa.min-saude.pt (H.D.); francisca.coutinho@insa.min-saude.pt (M.F.C.); luciana.moreira@insa.min-saude.pt (L.M.); 2Center for the Study of Animal Science-Instituto de Ciências, Tecnologias e Agroambiente da Universidade do Porto, CECA-ICETA, University of Porto, Praça Gomes Teixeira, Apartado 55142, 4051-401 Porto, Portugal; 3Associate Laboratory for Animal and Veterinary Sciences, AL4AnimalS, Faculdade de Medicina Veterinária Avenida da Universidade Técnica, 1300-477 Lisboa, Portugal; 4Biology Department, Faculty of Sciences, University of Porto, Rua do Campo Alegre, 4169-007 Porto, Portugal

**Keywords:** microRNA, lipid storage disorders, metabolism, cholesterol trafficking, biomarkers, Niemann-Pick type C, RNA-based therapies

## Abstract

Lipids are essential for cellular function and are tightly controlled at the transcriptional and post-transcriptional levels. Dysregulation of these pathways is associated with vascular diseases, diabetes, cancer, and several inherited metabolic disorders. MicroRNAs (miRNAs), in particular, are a family of post-transcriptional gene repressors associated with the regulation of many genes that encode proteins involved in multiple lipid metabolism pathways, thereby influencing their homeostasis. Thus, this class of non-coding RNAs (ncRNAs) has emerged as a promising therapeutic target for the treatment of lipid-related metabolic alterations. Most of these miRNAs act at an intracellular level, but in the past few years, a role for miRNAs as intercellular signaling molecules has also been uncovered since they can be transported in bodily fluids and used as potential biomarkers of lipid metabolic alterations. In this review, we point out the current knowledge on the miRNA signature in a lysosomal storage disorder associated with lipid dysfunction, Niemann-Pick type C, and discuss the potential use of miRNAs as biomarkers and therapeutic targets for RNA-based therapies.

## 1. Introduction

Lipids are key molecules not only for cellular membrane integrity and tissue physiology but also for energy storage and signaling. Phospholipids, glycolipids, and cholesterol constitute the major kinds of membrane lipids [1], and detrimental changes in lipid metabolism contribute to a variety of metabolic disorders, such as diabetes, obesity, liver disease, and atherosclerosis. Less frequent diseases related to lipid dysfunction include a subgroup of lysosomal storage disorders (LSDs). Lysosomal lipid storage disorders are rare inherited metabolic diseases that, among other mechanisms, cause the dysregulation of lipid metabolism and the accumulation of a broad range of lipids inside lysosomes, resulting in cellular toxicity [2,3,4]. Neurons are particularly sensitive to lipid accumulation, which leads to stunted brain development and neurodegeneration [5]. Over time, this excessive storage of fats can cause permanent cellular and tissue damage in the peripheral nervous system, liver, spleen, and bone marrow. Many of these monogenic diseases are fatal at a young age, and treatment options are limited [6]. The diagnostic timeline is generally very prolonged due to phenotypic overlapping and clinical heterogeneity [7,8,9,10], which may happen even in monozygotic twins [11]. Furthermore, response to therapy, when existing, can be very variable among patients with the same genotype and even within the same family. That is why the traditional monogenic view may not be enough to explain LSD pathophysiology, and classical genotype/phenotype correlations may not be as straightforward as previously thought.

Recent genomic approaches have contributed to the understanding of the complexity and clinical variability observed in this group of monogenic disorders, namely epigenetic studies [12,13,14,15,16]. Overall, the field of epigenetics is gathering momentum, and correlating epigenetic changes with clinical parameters is a commonly used approach these days. Among the broad field of epigenetics, microRNA (miRNA) profiling holds great promise to identify biomarkers that correlate with disease severity and assist in monitoring disease progression and the effectiveness of therapies [13]. Ultimately, miRNAs can also be the target of RNA-based therapies.

miRNAs are a class of highly conserved [17,18,19] single-stranded non-coding RNA (ncRNA) molecules with 20–22 nucleotides. They direct the post-transcriptional regulation of target messenger RNAs (mRNAs) in diverse eukaryotic lineages [18], fine-tuning their levels. Each miRNA can target multiple mRNAs that contain the cognate miRNA-binding site, thus being able to regulate multiple cellular functions involved in disease development and progression—including metabolism [20]. Similarly, each single miRNA can target multiple effectors of pathways in a functional gene network. In most cases, miRNAs suppress gene expression by interacting with the 3′ untranslated region (UTR) of target mRNAs and repress protein production by destabilizing the mRNA and via translational silencing [21,22]. Although most miRNAs act in the tissue where they are produced, some miRNAs can also be secreted from cells into the circulatory system [23]. Those circulating miRNAs are disease-specific miRNAs that may also contribute to the development of precision treatments. In rare diseases, however, these studies have been mostly underexploited. Some recent reports related to changes in miRNA expression and their consequences on LSD physiopathology have been described for Gaucher (type I, MIM #230800) [24] and Fabry diseases (MIM #301500) [25]. Here, we summarize the current knowledge of the miRNA profile of another lipid LSD—Niemann-Pick type C (NPC, MIM #257220)—and discuss the identification of dysregulated miRNAs in NPC patients that correlate with their high clinical heterogeneity and their potential use as potential diagnostic and/or prognostic biomarkers for NPC. We end on a brief note regarding the promising emerging ncRNA therapeutic approaches.

## 2. MicroRNA (miRNA) Biogenesis

The biogenesis of miRNAs can be classified into canonical and non-canonical pathways [22]. Overall, miRNA biosynthesis pathways are under tight temporal and spatial control, and their dysregulation is associated with many human diseases [26].

Currently, there are 576 *bona fide* miRNA genes annotated in the human genome [27,28], with a significant proportion exhibiting non-ubiquitous, cell type-specific expression patterns [29]. These genes range from independent transcription units (intergenic miRNAs) to intron-contained, protein-coding gene-associated genomic *loci* (intragenic miRNAs). In humans, most miRNAs are intragenic [30] and located within introns [31], often being functionally connected to their host genes as a gene network [32].

The canonical biogenesis of miRNAs, depicted in Figure 1, begins in the nucleus, where both intergenic and intragenic miRNAs are transcribed by RNA polymerase II into long primary miRNA transcripts (pri-miRNAs) [33] before being cleaved by the Drosha ribonuclease (RNase) III enzyme and its interacting protein DGCR8, to generate a hairpin-structure where miRNAs are embedded (pre-miRNA). This molecule is then exported to the cytoplasm via the Exportin5 (XPO-5)-RanGTP system [26], where it is then cleaved by the Dicer RNase III enzyme and its interacting protein TRBP, releasing a ~22 nt mature miRNA duplex (miRNA-3p/miRNA-5p), which is loaded onto a member of the Argonaute (AGO) protein family to form the RNA-induced silencing complex (RISC) [26,34,35]. Typically, one particular strand of the miRNA duplex is favored by the RISC, while the other is ejected from the complex and degraded [26,36,37]. miRNAs often exert their effect by downregulating mRNA levels, but they may also directly repress the translation of hundreds of genes [38,39,40], fine-tuning protein synthesis. One such example is their ability to target components of the epigenetic machinery such as DNA methyltransferases (DNMTs), ten-eleven translocation (TET) proteins, histone deacetylases (HDACs) and histone methyltransferases (HMTs) [41], thus indirectly regulating DNA methylation and histone modifications.

Apart from the well-known canonical pathway, non-canonical pathways for miRNA biogenesis are also emerging, which can be generally grouped into Drosha/DGCR8- and Dicer-independent pathways [22,34]. Examples of pre-miRNAs generated by these independent pathways include mirtrons—which are produced from mRNA introns during splicing—and the 7-methylguanosine (m7G)-capped pre-miRNA, and directly exported to the cytoplasm via Exportin 1 without the need for Drosha cleavage [22].

## 3. miRNAs and Lipid Homeostasis

The liver plays a crucial role in the metabolism of most endogenous lipids, plasma apolipoproteins, and lipoproteins, keeping lipid biogenesis and turnover carefully controlled by regulating the distribution of lipoprotein particles within the body. Low-density lipoprotein (LDL) particles are synthesized mainly in hepatocytes and transport cholesterol, triglycerides, and phospholipids to cells in the periphery. Hepatocytes are also the main source of high-density lipoprotein (HDL) particles released into circulation. The balance of LDL and HDL in circulation is critical for lipid homeostasis, namely that of cholesterol. Imbalances in the cellular cholesterol concentration promote vascular diseases, diabetes, and cancer but can also be observed in several monogenic diseases [42]. In fact, cholesterol homeostasis is tightly modulated by a complex network, which involves its synthesis, import, export, esterification, and metabolism. This balance is maintained by two key transcription factors: the membrane-bound transcription factor—sterol regulatory element-binding proteins SREBP1 and SREBP2 [43]—and the liver X receptor (LXR) [44], that activate genes encoding enzymes involved in cholesterol and fatty acid biosynthesis. The LXR—an oxysterol-activated transcription factor—regulates the expression of genes involved in the efflux of cholesterol, such as the adenosine triphosphate-binding cassette (ABC) transporters ABCA1 and ABCG1. Extracellular cholesterol absorption and distribution into cells requires an appropriate endosomal trafficking system. LDL binds to its receptor on the cell surface and is then absorbed by clathrin-mediated endocytosis. Upon internalization, LDL is delivered to early sorting endosomes and then to late endolysosomes/lysosomes, where LDL and cholesteryl esters are hydrolyzed. The LDL receptor can then be recycled back to the plasma membrane. After the hydrolyzation of cholesteryl esters by lysosomal acid lipase (LAL), the lysosomal Niemann-Pick type C1 and C2 proteins mediate cholesterol export to the cytosol. The small soluble NPC2 protein picks up cholesterol inside late endosomes/lysosomes and delivers it to NPC1—a lysosomal membrane protein—which is then thought to transport cholesterol into and/or across the lysosomal membrane [45,46,47,48,49]. The mechanism through which cholesterol is delivered to other membranes, such as those from the ER, endosomes, mitochondria, and the plasma membrane, is still largely unknown.

In addition to classic transcriptional regulation of cholesterol metabolism, several miRNAs have been identified to be potent post-transcriptional regulators of lipid metabolism genes, including let-7, miR-26, miR-33, miR-34a, miR-106, miR-122, miR-143, miR-148-a, miR-185 [50], miR-335, miR-370, miR-370, miR-378/378*, and miR-758 [44,51,52,53].

## 4. Niemann–Pick Type C Disease

Pathogenic variants in the *NPC1* or *NPC2* genes lead to cellular cholesterol trafficking impairment, triggering NPC disease. Additionally, *NPC1* gene variants are implicated in different common metabolic disorders, an observation that can be explained using NPC1 protein influence on steroid hormone production and/or lipid homeostasis [54]. Clinically, NPC is a devastating neurodegenerative disease with a wide range of phenotypes and differing ages of onset. Patients who develop NPC during early infancy frequently present mainly visceral manifestations such as splenomegaly, hepatomegaly, neonatal jaundice, and hyperbilirubinemia, with several degrees of neurologic signs [55,56]. Adolescent or adult-onset NPC patients, on the other hand, display varying combinations of progressive neurologic deficits, e.g., ataxia, dystonia, and/or dementia, vertical supranuclear gaze palsy (VSGP), or major psychiatric illness [56].

Classically, NPC diagnosis relied upon filipin staining of cultured fibroblasts from skin biopsies [57]. Filipin specifically binds to unsterified cholesterol, allowing the cholesterol accumulated in perinuclear vesicular compartments to be visualized [58]. That is why Filipin labeling was used for years as the gold standard for NPC diagnosis. Nevertheless, even the most severely affected patients may fail to be diagnosed using this method [59,60]. Recent advances in the field, such as the generalized use of next-generation sequencing (NGS) technologies, namely whole-exome sequencing (WES) or whole-genome sequencing (WGS), have been actively increasing the detection of NPC patients. The development of novel, rapid, and reliable biomarkers has also greatly contributed to this scenario. Among others, a biomarker with great potential to shorten the diagnostic odyssey of NPC patients is N-palmitoyl-O-phosphocholineserine (previously known as lysosphingomyelin-509), which has been shown to be elevated in plasma and dried blood spots of NPC patients [61,62].

Other studies have highlighted proteins as potential biomarkers, including GPNMB, that corresponds to the glycoprotein NMB, encoded by the *GPNMB* gene—a downstream target of the lysosomal transcription factor EB—TFEB. GPNMB was recently identified as a useful biomarker of treatment response in NPC disease [63]. Besides biochemical biomarkers, other kinds of blood biomarkers, particularly miRNAs, are already used for cancer and other diseases and should be explored for NPC.

### 4.1. MicroRNA Profile in NPC

Besides their putative use as biomarkers, knowledge of miRNA signatures can be useful in understanding the different clinical manifestations of NPC. Like in many other LSDs, a strictly monogenic view is not enough to explain the significant clinical variability that occurs in this disease, and that may be seen even within members of the same family. Over the following paragraphs, we will review the current knowledge on the miRNA profile in NPC by summarizing the currently published results reported by different teams.

A previous study of miRNA profile was conducted in NPC skin fibroblasts using microarrays (Human microRNA Panel A), encompassing 365 mature human miRNAs, even though only 107 of them were shown to be expressed in NPC fibroblasts samples [64]. No information about the genotype was provided. Differences in gene expression between control and NPC fibroblasts (>3.5-fold difference, *p*-value < 0.05) were detected, and it was found that three miRNAs (miR-196a, miR-196b, and miR-296) were significantly upregulated, whereas 38 miRNAs (35.5%) were significantly downregulated in NPC cells (Table 1). The predicted target genes of the differentially expressed miRNAs were mostly related to the regulation of transcription, signal transduction, and vesicular transport, and only a small percentage was linked to cholesterol and lipid metabolism. Amongst the downregulated non-coding RNAs, miR-98 was the most downregulated, and its predicted target genes were found to be related to the lysosome and Golgi. miR-143, a lipid biosynthesis-associated miRNA, had a 20-fold decreased expression in NPC cells. The most upregulated one (near forty-fold) was miR-196a, which, as well as miR-98, was also found to be related to the lysosome and Golgi compartments in NPC cells. Altogether, the technology used in this study presents some caveats since microarrays have some well-known technical limitations. Furthermore, the predicted target genes of the differentially expressed miRNAs were not experimentally validated. Still, this is preliminary data, which surely deserves further validation. More recently, in the era of NGS-based methods, RNA sequencing (RNA-Seq) technologies allow a higher coverage and offer higher resolution, as well as a better range of detection and lower technical variability when compared with microarrays [65], and it is expected to find more dysregulated miRNAs in NPC patients’ cells using RNA-Seq based approaches, namely ncRNA pipelines.

Another study regarding miRNAs in NPC disease started with a proteomics analysis on an NPC-null mouse model (*Npc1*^−/−^ mice) [66], which mimics human disease and presents progressive cerebellar neurodegeneration, ataxia, and hepatosplenomegaly. This study was conducted using liver and spleen samples, two of the organs that are strongly affected in the early infantile form of the NPC disease. Neurodegeneration has been studied more extensively, but severe liver disease is also a major symptom for a subset of patients and some of them died from acute liver failure by the first months of life [67,68]. In this context, capillary LC-MS/MS was used to perform label-free quantitative proteomics in *Npc1*^−/−^ mice at an endpoint in disease progression. The differential proteome was analyzed to identify potential biomarkers and to unravel the mechanism leading to hepatosplenomegaly. Overall, the analysis of the datasets showed many altered pathways in the *Npc1*^−/−^ mice and upstream transcriptional regulator analysis identified miR-155 as a possible modulator. For validation, the expression levels of miR-155 were quantified by qRT-PCR in 9-week-old *Npc1*^+/+^ and *Npc1*^−/−^ mice and its levels were found to be significantly decreased in the spleen and increased in the liver of *Npc1*^−/−^ mice [66]. This miRNA—which is involved in the regulation of multiple genes involved in lipid metabolism—was thereby proposed as a novel indicator of spleen and liver pathology in NPC disease.

Further, miR-155 enhances tyrosine-kinase-mediated signaling pathways in the immune system, playing a role in inflammation, which is a hallmark of NPC disease [69,70]. In fact, miR-155 targets the SHIP tyrosine phosphatase [71]—a well-known negative regulator of PI3P-AKT signaling. It has been described that tyrosine kinase levels are dysregulated in NPC neurons from transgenic mice, for instance [72]. Therefore, tyrosine-mediated signaling is enhanced in these cells. The aberrant activation of certain tyrosine kinases leads to premature neuroinflammation and neuronal loss in the forebrain of NPC transgenic mice [72].

Whether these differentially expressed intracellular miRNAs (detected in fibroblasts, liver, or spleen cells) could be used as biomarkers as a result of their association with extracellular vesicles (EVs) needs to be studied further. On the other hand, investigating if the expression of those miRNAs varies even among NPC patients with the same pathogenic variant in the *NPC1* gene could explain the large phenotypic spectrum characteristic of these patients.

### 4.2. miRNAs That Target the NPC1 Gene, Impacting Its Expression

Besides the miRNA signature associated with NPC disease, another point of interest in the scope of this review is to explore the miRNAs that regulate NPC1 levels. To the best of our knowledge, miR-33a/b, miR-25, and miR-200c bind to the 3′-UTR of the *NPC1* mRNA, thus repressing its expression (Figure 2). Over the following paragraphs, we will further elaborate on each one of these miRNAs and the original studies reporting their impact on the expression of NPC1.

#### 4.2.1. miR-33a/b

Among the most well-characterized key regulators of lipid metabolism are miR-33a and -b (miR-33 a/b) [73,74]—which are encoded within the introns of *SREBP2* and *SREBP1*, respectively—and constitute a master switch for genes involved in fatty acid and cholesterol metabolism [73,75,76,77,78,79] and insulin regulation. miR-33a and miR-33b share the same seed sequence and differ in only two nucleotides. Both regulate cholesterol homeostasis and fatty acid metabolism in concert with their host genes, *SREBP2* and *SREBP1c*, in a negative feedback loop [80]. *SREBP1* encodes SREBP1a and SREBP1c, which regulate lipogenic genes, whereas *SREBP2* mainly modulates cholesterol-regulating genes.

miR-33a specifically targets the 3′-UTR of the *NPC1* mRNA, among others, thus playing an important role in cholesterol homeostasis. The 3′-UTR of human *NPC1* contains two miR-33a binding sites, resulting in the repression of NPC1 protein in hepatocytes and macrophages [76]. For example, under low intracellular cholesterol conditions, miR-33a is cotranscribed with SREBP2 and works to increase the cellular cholesterol levels via the inhibition of ABCA1 and ABCG1 transporters and NPC1 protein [73,75,76], thus reducing cholesterol export. On the other hand, in cholesterol-rich conditions, SREBP2 and miR-33a levels decreases, thus relieving ABCA1 suppression [80]. When the intracellular cholesterol levels are high, SREBP1-c is induced, miR-33b is co-transcribed and works to reduce cellular fatty acid oxidation by targeting carnitine O-octanoyl transferase (CROT), carnitine palmitoyltransferase 1A (CPT1a), hydroxyacyl coenzyme A (hydroxyacyl-CoA) dehydrogenase subunit beta (HADHB), and AMP kinase subunit alpha (AMPKα), as well as insulin signaling, by targeting insulin receptor substrate 2 (IRS2) and sirtuin 6 (SIRT6) [81].

In conclusion, SREBP’s transcription factors, in concert with miR33a/b, prevent dramatic increases or reductions in cellular lipid levels by controlling lipid synthesis, uptake, and export. miR-33a and miR-33b help prevent further loss of lipids by targeting genes involved in cholesterol trafficking and efflux (*NPC1*, *ABCA1*, *ABCG1*) and fatty acid metabolism (*CPT1A*, *CROT*).

#### 4.2.2. miR-25

miR-25 has a broad range of potential mRNA targets that participate in many cellular processes. Among those, autophagy plays an essential role, as it degrades cytoplasmic constituents and organelles, strongly contributing to the cell’s homeostasis. Additionally, it was precisely the pivotal role of miR-25 in autophagy that was first brought to light. In fact, while studying miRNA effects on autophagy regulation during the progression of *Mycobacterium tuberculosis* (Mtb)/ Bacillus Calmette-Guérin (BCG) infection in macrophages, some miRNAs were identified as upregulated, including miR-25. Then, online software (TargetScan https://www.targetscan.org/vert_80/, accessed on 21 August 2023. miRDB http://www.mirdb.org/, miRanda https://bioweb.pasteur.fr/packages/pack@miRanda@3.3a, accessed on 21 August 2023) was used to predict the mRNA targets of miR-25, and those results were subsequently cross-linked with target autophagy pathway-related genes [82,83]. The results showed that miR-25—which is upregulated upon BCG/Mtb infection—directly targets the *NPC1* mRNA (binding the 3′-UTR), decreasing NPC1 at the transcriptional and protein levels and impairing autophagosome-lysosome fusion [84]. Accordingly, when the cells were transfected with a miR-25 inhibitor, NPC1 was upregulated, promoting autophagic flux. Ultimately, all the results suggested that miR-25 actively affects the intracellular survival of BCG and Mtb by regulating the autophagic pathway via its target—NPC1.

#### 4.2.3. miR-200c

As we have already mentioned, miR-25 is not the only miRNA known to directly bind *NPC1* mRNA. miR-200c has also been shown to play its part. Additionally, again, the first reports on miR-200c, came from studies in cancer cells. It was initially shown to inhibit epithelial-mesenchymal transition (EMT)—a mechanism that facilitates metastasis in cancer cells. In order to unveil the genes affected by miR-200c, it was observed that the restoration of miR-200c to cancer cells reduces the levels of certain immune modulatory response genes and among them, *NPC1* was found [85]. Consistently, its silencing led to decelerated proliferation and decreased invasiveness, altered mitochondrial function and morphology, suppression of mTOR signaling and accumulation of autophagossomes [86]. Additionally, the *NPC1* gene has also been shown to be highly expressed and directly targeted by miR-200c in aggressive breast cancer [86].

## 5. Circulating miRNA as Putative Biomarkers

Extracellular miRNAs are remarkably stable, and since they can be used in intercellular communication [87], their characterization can provide important insights into health and disease. For neurodegenerative and metabolic diseases, such as NPC, it is expected to find informative miRNAs in bodily fluids, which can be used as reliable biomarkers for disease diagnosis and prognosis. The possibility of using miRNAs as disease biomarkers and treatment follow-ups has already been studied for another LSD, namely Pompe Disease (Glycogen storage disease type II: MIM #232300) [88].

Cell-free miRNAs can be transported using two tightly regulated and energy-demanding processes (Figure 3). The first one is active transport via EVs, which includes shedding vesicles/microvesicles/ectosomes and exosomes [23], which are released by almost all cell types under physiological and pathological conditions [89]. The second is transport as part of protein-miRNA complexes (RNA-binding proteins) (Figure 3). In some cases, there can also be passive leakage of miRNAs from broken and damaged cell membranes. For the uptake of circulating miRNAs, microvesicle (MV)-enclosed miRNAs are recognized by cellular surface molecules and internalized by endocytosis, phagocytosis, or direct fusion with the plasma membrane.

Major cell types of the central nervous system (CNS) release EVs, namely exosomes. It was shown that patients harboring pathogenic NPC1 variants showed elevated exosomal cholesterol release, which could disrupt cholesterol homeostasis [90].

## 6. Therapeutic Metabolic Reprogramming to Use in LSDs—RNA-Based Therapeutics

Finally, there is yet another possibility to consider when discussing miRNAs: their potential as therapeutic targets for RNA-based therapies. In fact, a better understanding of the epigenome, especially regarding the role of miRNAs and their expression levels, namely if some of them are up or downregulated in NPC patients, can also lead to the development of tailored therapies. One possibility is the use of RNA-based therapeutics, a type of therapy that has merited significant attention in recent years due to their potential to treat a variety of diseases [91,92].

The most well-known RNA-based therapeutics rely on the use of modified oligonucleotides, also known as antisense oligonucleotides (ASOs) [93]. ASOs are single-stranded oligonucleotides, some of which can also be used for target gene knockdown via the recruitment of endogenous RNAse H (gapmer ASOs). However, another type of ASO, designated steric block ASOs, exerts its function by binding to their target transcripts, thus masking specific sequences and interfering with RNA–RNA and/or RNA–protein interactions [93,94].

Steric block ASOs can either be used to competitively inhibit miRNAs via direct binding to these small RNA species (in which case they are known as anti-miRNA oligonucleotides, anti-miRs, or antagomirs), or they can be designed to mimic the activity of endogenous miRNAs (being referred to as miRNA mimics or agomirs) [93].

Importantly, these types of miRNA-targeting ASOs are under clinical development for a number of diseases, with ongoing clinical trials and some approved drugs, which further support their overall potential as effective therapeutic strategies as soon as their safety and toxicology profiles meet all the requirements. The first anti-miRNA drug to enter clinical trials was miravirsen—an antagomir to miR-122, a liver-specific microRNA that the hepatitis C virus (HCV) requires for replication. Despite having shown promising results, an escape mutation that renders the HCV genome refractory to miravirsen has also been reported [94,95].

However, many other approaches have entered development either to restore or to block miRNAs for a therapeutic purpose. Anti-miRNA oligonucleotide drugs are being developed for nonalcoholic steatohepatitis, cholestatic diseases, fibrogenesis of organs associated with Alport syndrome, lymphoma and leukemia, amyotrophic lateral sclerosis, and ischemic conditions. In turn, miRNA mimic approaches are being developed and are currently in clinical trials for fibrotic diseases, cancers, and malignant pleural mesothelioma [95].

To the best of our knowledge, the use of oligonucleotide drugs to modulate the function of miRNAs has never been tested, not even *in vitro*, for NPC or any other LSD in which certain miRNAs were found to be differentially expressed.

However, another type of RNA therapeutics that is based on the use of the entire coding mRNA molecule has been recently tested *in vitro* for NPC. Furtado et al. treated NPC patient fibroblasts with an engineered *NPC1* mRNA and observed the restoration of functional protein expression and the reversal of disease pathology [96]. Therefore, RNA therapies hold promise for the treatment of NPC, and they can be used to target both the genetic causes (mutations in the *NPC1* and *NPC2* genes) as well as the epigenetic causes, such as the dysregulation of miRNAs.

In summary, even though many issues still have to be addressed before the broader application of miRNA-based therapies (e.g., delivery, off-targets, immune system reactions), their potential for the treatment of a great panoply of diseases in which microRNAs are dysregulated (NPC and other LSDs included) is enormous.

## 7. Concluding Remarks

NPC is a neurovisceral disease, which means its main affected organs are all of great interest for targeted analyses since miRNA expression is cell-type specific. That catalog includes not only the brain but also the liver and spleen, just to mention the most significant ones. However, the few studies reported so far did not cover all of those major organs. Furthermore, most of them were assessed using microarray technologies, which have quite a few limitations. Recent advances in research, including the development of multi-omics methods, have made it possible to elucidate multiple transcriptomic and epigenetic pathways in eukaryotic organisms. Decreased sequencing costs and increased accessibility of data analysis make epigenetics research more affordable and achievable. For monogenic diseases with high clinical variability, such as LSDs in general and NPC in particular, epigenetics, namely ncRNAs, can explain the different disease outcomes observed in patients with the same genetic defect. In fact, it is predicted that miRNAs account for 1–5% of the human genome but regulate at least 30% of protein-coding gene expression, and they are not limited to intracellular activity. Due to their ability to be transported in EVs, their detection in bodily fluids can provide important insights related to the disease and be used as biomarkers.

This review highlighted the microRNA profile associated with NPC and some of the few miRNAs that have been experimentally shown to target the *NPC1* gene, thus modulating relevant cellular mechanisms. However, most importantly, it also highlighted the need for additional studies and extended profiling in NPC patients. Overall, the potential of miRNAs is enormous, not just from a basic science point of view (by helping us unveil and understand the underlying pathophysiological mechanisms) but also from a more applied perspective (by acting as putative disease biomarkers or therapeutic targets). Either way, there is plenty of room and need for further research in the field.

## Figures and Tables

**Figure 1 biomedicines-11-02615-f001:**
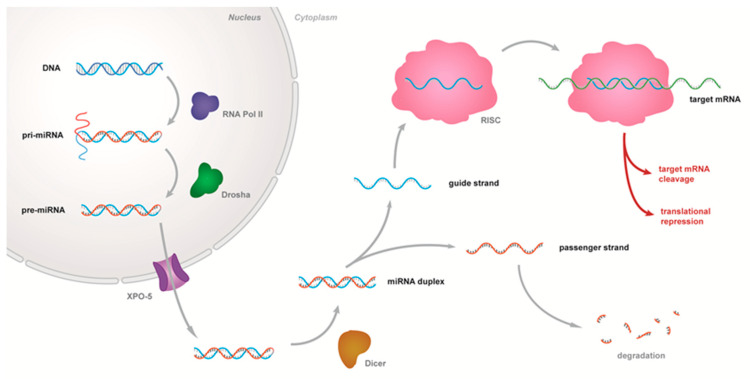
Canonical miRNA biosynthesis. MicroRNAs (miRNAs) function as guide molecules in RNA silencing. In animals, miRNAs are ~22 nucleotides in length, and they are processed by the sequential action of two RNase III enzymes—Drosha and Dicer. miRNA biogenesis is regulated at multiple levels, including at the transcriptional level and upon their cleavage by Drosha and Dicer in the nucleus and cytoplasm, respectively.

**Figure 2 biomedicines-11-02615-f002:**
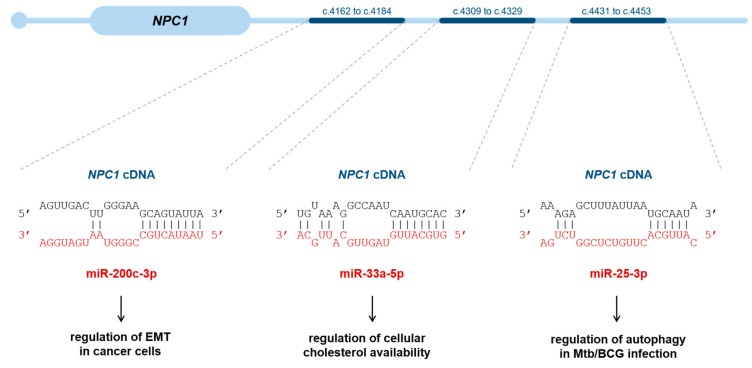
Different miRNAs target the *NPC1* gene under different physiological contexts. miR-200c-3p, a known repressor of EMT in cancer cells, binds to the -3′UTR of *NPC1;* miR-33a-5p targets the *NPC1* gene, regulating the cellular cholesterol availability, and miR-25-3p also represses *NPC1* expression, regulating autophagy in Mtb/BCG infection.

**Figure 3 biomedicines-11-02615-f003:**
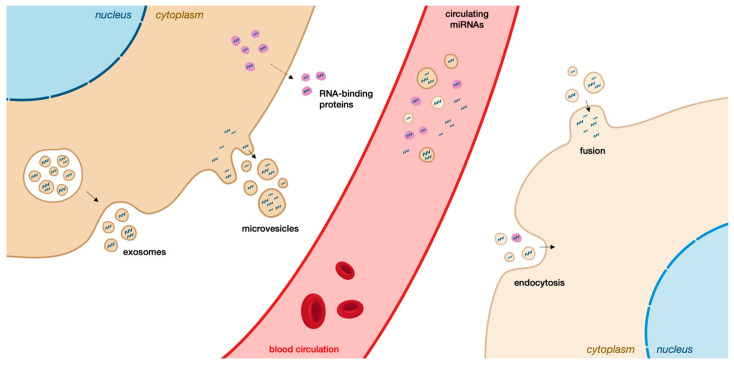
miRNAs Export, transport, and uptake. Extracellular microRNA can be transported in a regulated manner through exosomes (30–100 nm), microvesicles (100–1000 nm), and complexed with proteins. Circulating miRNAs can be used as biomarkers and function in intercellular communication.

**Table 1 biomedicines-11-02615-t001:** Most differentially expressed miRNAs in NPC cells (human and *Npc1*^−/−^ mice).

miRNA	RQ Value ^1^ (NPC/Control)	Expression	Tissue of Origin	Organism	Technology Applied	Ref
miR-98	−33.3	Decreased	Skin fibroblasts	NPC patients	TaqMan^®^ low-density arrays	[64]
miR-143	−20	Decreased				
miR-196a	37.41	Increased				
miR-196b	6.68	Increased				
miR-296	3.53	Increased				
miR-155	n/a	Increased	Liver			
	n/a	Decreased	Spleen	*Npc1*^−/−^ mice	qRT-PCR	[66]

^1^ RQ = Relative quantity value.

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
