# Peer review of "MicroRNA Profile, Putative Diagnostic Biomarkers and RNA-Based Therapies in the Inherited Lipid Storage Disease Niemann-Pick Type C"

_biomedicines, 2023, doi:10.3390/biomedicines11102615_

Round 1
Reviewer 1 Report
Dear the Editor
Encarnacao M et al reviewed recent topics on microRNA involved in Niemann-Pick disease type C (NPC). These authors discussed several selected microRNAs with an altered expression in NPCs as a potential clinical biomarker. For further extension of RNA-based therapy, antisense oligonucleotide-based technology was briefly described. Overall, this is a concise mini-review summarizing previous literatures, although unfortunately most of them were published before 2018.
Major concerns:
1) In section 6, is there any study of RNA-based therapy for NPC?
2) In section 4, L175-, there is a good diagnostic biomarker for NPC (PMID: 31658747; 31201291).
3) In section 4.1, L231, miR-155 is known to enhance tyrosine kinase-mediated signaling pathway in immune system. Please comment the connection between this and pathogenesis of NPC.
Minor concerns:
1) L290, there is a typo. Please check entire manuscript again.
2) L202, 212, and others, the Reviewer felt that DE (differentially expressed) appeared to be better spelt out, but not to be abbreviated.
Author Response
Major concerns:
1) In section 6, is there any study of RNA-based therapy for NPC?
In this section, we detail only those RNA therapies that are likely to target differentially expressed miRNAs and that rely on the use of oligonucleotide drugs. However, another type of RNA therapy has recently been tested in vitro for NPC and so we have added a new paragraph mentioning that study (L375 to L384 in the revised version).
2) In section 4, L175-, there is a good diagnostic biomarker for NPC (PMID: 31658747; 31201291).
In section 4, L175 of the submitted version, we discuss the former method for NPC diagnosis and comment that none of the available biomarkers are truly specific for NPC. Although it is published that the compound known as Lyso-SM-509 (identified as palmitoyl-O-phosphocholineserine) is elevated in NPC patients’ plasma, our experience says otherwise (unpublished). We have analyzed a few cases of NPC patients with no significant elevation of Lyso-SM-509 in plasma, compared to controls.
As suggested by the reviewer, we edited the text to mention the Lyso-SM-509 biomarker and the respective publications (L174 to L177 in the revised version). We also included information about other putative biomarkers, mainly to monitor the treatment response. In fact, additional studies have highlighted proteins as potential biomarkers, including the transmembrane glycoprotein NMB (encoded by the GPNMB gene). This gene is a downstream target of the transcription factor EB (TFEB). In that study, immunohistochemistry of the cerebellum as well as measurements of plasma from Npc1m1N null mice treated with HPβCD (cyclodextrin) and adeno-associated virus gene therapy suggests that the identified gene, GPNMB, may serve as a useful biomarker of treatment response in NPC1 disease (this information was included in L178 to L181 in the revised version). We also believe that differentially expressed microRNAs can be used as good biomarkers for diagnosis and good therapeutic targets for future therapies, namely RNA-based therapies, as discussed in this review.
3) In section 4.1, L231, miR-155 is known to enhance tyrosine kinase-mediated signaling pathway in immune system. Please comment the connection between this and pathogenesis of NPC.
Indeed, miR-155 targets the inositol phosphatase SHIP1, which contains an SH2 domain and is expressed mainly in hematopoietic cells, but also in microglia. SHIP1 inhibits the tyrosine kinase-mediated signaling pathway in the immune system and, as miR-155 targets SHIP1, by downregulating it, tyrosine-mediated signaling is enhanced/activated in NPC neurons, for instance. Aberrant activation of certain tyrosine kinases leads to early neuroinflammation and loss of neurons in the forebrain of Niemann–Pick type C (NPC) transgenic mice. This information was included in the new version of the manuscript (L231 to L238 of the revised version).
Minor concerns:
1) L290, there is a typo. Please check entire manuscript again.
The entire manuscript was checked for typos.
2) L202, 212, and others, the Reviewer felt that DE (differentially expressed) appeared to be better spelt out, but not to be abbreviated.
The DE abbreviation was removed and spelt out.
Reviewer 2 Report
The manuscript by Ernação and coauthors focuses on the description of miRNAs that are involved in Niemann-Pick type C disease.
The field is interesting both for understanding the pathophysiology of the disease and for developing new innovative therapeutic treatments.
However, the review at this stage is still preliminary. I suggest the authors significantly revise the manuscript.
Main points:
1- Authors must reorganize the text avoiding repetitions (for example, the characteristics of miRNAs are in the introduction and in paragraph 2) and deleting or modifying some paragraphs that are not the focus of the work. For example, in paragraph 3, only the last sentence referred to the paragraph title. Furthermore, section 6 discusses the relevance of miRNAs for therapeutic applications, but the correlation with Niemann-Pick type C disease is not well described.
2- Figure 2 needs to be improved. Authors should indicate the nucleotide positions involved in the coupling interaction.
3- All sections of the text could be improved. In particular, the role of mir33a/b needs to be better documented and described
Author Response
Main points:
1. Authors must reorganize the text avoiding repetitions (for example, the characteristics of miRNAs are in the introduction and in paragraph 2) and deleting or modifying some paragraphs that are not the focus of the work. For example, in paragraph 3, only the last sentence referred to the paragraph title. Furthermore, section 6 discusses the relevance of miRNAs for therapeutic applications, but the correlation with Niemann-Pick type C disease is not well described.
In the revised version, the text was reorganized and repetitions were avoided, namely in “1. Introduction and 2. MicroRNA (miRNA) Biogenesis”
Concerning section 6, this section was included in the manuscript due to the high potential of these therapies to correct the pathological effects of differentially expressed miRNAs. Since differentially expressed miRNAs are being described in NPC and other LSDs, we think this section adds value to the review and can be used in the future as a basis for developing new therapies for these diseases.
However, we agree with the referee that the connection with NPC was not very clear and we decided to withdrawn the information that is not as relevant in this context, namely the description of siRNAs or and the reference to the new antagomir for HCV infection.
In turn, to make the connection with NPC we added the following paragraph:
“To our knowledge, the use of oligonucleotide drugs to modulate the function of miRNAs has not yet been tested, even in vitro, for NPC1 or for any other LSD in which miRNAs were found to be downregulated (L375 to L377 in the revised version).
However, another type of RNA therapeutics that is based on the use of the entire coding mRNA molecule has been recently tested in vitro as a potential therapeutic approach for NPC1. Furtado et al treated NPC1 patient fibroblasts with an engineered NPC1 mRNA and observed the restoration of functional protein expression and reversal of disease pathology. Therefore, RNA therapies hold promise for the treatment of NPC and can be used to target both the genetic causes (mutations in the NPC1 and NPC2 genes) as well as the epigenetic causes such as dysregulation of miRNAs, which are the main focus of this review.”
2- Figure 2 needs to be improved. Authors should indicate the nucleotide positions involved in the coupling interaction.
Figure 2 was improved and the nucleotide positions were added, as suggested.
3.All sections of the text could be improved. In particular, the role of mir33a/b needs to be better documented and described
More information regarding the role of miR-33a/b was added to the manuscript and the whole section was reformulated (lines 255 to 281 in the revised version).
Round 2
Reviewer 1 Report
Dear the Editor
All concerns have been reasonably cleared by this revision.
Reviewer 2 Report
no comments